# Abundances, emissions, and loss processes of the long-lived and potent greenhouse gas octafluorooxolane (octafluorotetrahydrofuran, $c$-C$_4$F$_8$O) in the atmosphere

Martin K. Vollmer[1], François Bernard[2,3,4], Blagoj Mitrevski[5], L. Paul Steele[5], Cathy M. Trudinger[5], Stefan Reimann[1], Ray L. Langenfelds[5], Paul B. Krummel[5], Paul J. Fraser[5], David M. Etheridge[5], Marc A. J. Curran[6,7], and James B. Burkholder[2]

[1]Laboratory for Air Pollution and Environmental Technology, Empa, Swiss Federal Laboratories for Materials Science and Technology, Dübendorf, Switzerland
[2]Earth System Research Laboratory, NOAA, National Oceanic and Atmospheric Administration, Boulder, Colorado, USA
[3]Cooperative Institute for Research in Environmental Sciences, University of Colorado, Boulder, Colorado, USA
[4]now at: Institut de Combustion Aérothermique, Réactivité et Environnement, Centre National de la Recherche Scientifique, Observatoire des Sciences de l'Univers en région Centre, Orléans, France
[5]Climate Science Centre, CSIRO Oceans and Atmosphere, Aspendale, Victoria, Australia
[6]Australian Antarctic Division, Kingston, Tasmania, Australia
[7]Antarctic Climate & Ecosystems Cooperative Research Centre, Hobart, Tasmania, Australia

*Correspondence to:* Martin K. Vollmer (martin.vollmer@empa.ch)

**Abstract.** The first observations of octafluorooxolane (octafluorotetrahydrofuran, $c$-C$_4$F$_8$O), a persistent greenhouse gas, in the atmosphere are reported. In addition, a complementary laboratory study of its most likely atmospheric loss processes, its infrared absorption spectrum, and global warming potential (GWP) are reported. First atmospheric measurements of $c$-C$_4$F$_8$O are provided from the Cape Grim Air Archive (41°S, Tasmania, Australia, 1978–present), supplemented by two firn air samples from Antarctica, in situ measurements of ambient air at Aspendale, Victoria (38°S), and a few archived air samples from the Northern Hemisphere. The atmospheric abundance in the Southern Hemisphere has monotonically grown over the past decades and leveled at 74 ppq (parts per quadrillion, femtomol mol$^{-1}$ in dry air) by 2015–2018. The growth rate of $c$-C$_4$F$_8$O has decreased from a maximum in 2004 of 4.0 ppq yr$^{-1}$ to <0.25 ppq yr$^{-1}$ in 2017 and 2018. Using a 12-box atmospheric transport model, globally averaged yearly emissions and abundances of $c$-C$_4$F$_8$O are calculated for 1951–2018. Emissions, which we speculate to derive predominantly from usage of $c$-C$_4$F$_8$O as a solvent in the semiconductor industry, peaked at 0.15 ($\pm$0.04, 2$\sigma$) kt yr$^{-1}$ in 2004 and have after declined to <0.015 kt yr$^{-1}$ in 2017 and 2018. Cumulative emissions over the full range of our record amount to 2.8 (2.4–3.3) kt, which correspond to 34 Mt of CO$_2$-equivalent emissions. Infrared and ultraviolet absorption spectra for $c$-C$_4$F$_8$O as well as the reactive channel rate coefficient for the O($^1$D) + $c$-C$_4$F$_8$O reaction were determined from laboratory studies. On the basis of these experiments, a radiative efficiency of 0.430 W m$^{-2}$ ppb$^{-1}$ (parts per billion, nanomol mol$^{-1}$) was determined, which is one of the largest found for synthetic greenhouse gases. The global annually averaged atmospheric lifetime, including mesospheric loss, is estimated to be >3 000 years. GWPs of 8 975, 12 000, and 16 000 are estimated for the 20, 100, and 500-year time-horizons, respectively.

## 1 Introduction

Halogenated organic substances are generally potent greenhouse gases and contribute significantly to climate change, despite their relatively low abundances in the atmosphere (Myhre et al., 2013; Carpenter and Reimann, 2014). Hydrofluorocarbons (HFCs), perfluorocarbons (PFCs), sulfur hexafluoride ($SF_6$), and nitrogen trifluoride ($NF_3$) are important anthropogenic green-house gases, which are included in the Kyoto Protocol to the United Nations' Framework Convention on Climate Change (UNFCCC). Although they do not have the capacity to destroy stratospheric ozone (unlike e.g. chlorofluorocarbons), HFCs have also been added to the Montreal Protocol on Substances That Deplete the Ozone Layer through the recent Kigali Amendment so that emissions can be curtailed by the effective method of restricting HFC use (United Nations, 2016).

The topic of the present research is the heterocyclic and fully fluorinated compound octafluorooxolane ($c$-$C_4F_8O$, CAS 773-14-8), better known by its older name as octafluorotetrahydrofuran, from which it has recently been renamed to its present name by IUPAC (Favre and Powell, 2014). The compound is listed in the Intergovernmental Panel on Climate Change (IPCC) 2006 guidelines in support of UNFCCC (IPCC (2006), Volume 1, Chapter 8) as a compound, for which countries are encouraged to provide emissions estimates (on a mass unit until a published greenhouse warming potential (GWP) will become available). In the 2013 Revisions of the UNFCCC reporting guidelines (UNFCCC, 2013), $c$-$C_4F_8O$ is absent from the list of compounds with mandatory reporting. Additional reporting regulations exist on country or state levels. For example in the USA, large suppliers and emitters of $c$-$C_4F_8O$ are required to report the amounts they supply or emit under the Greenhouse Gas Reporting Program (GHGRP, URL: https://www.epa.gov/ghgreporting, accessed January 2019). When $CO_2$-equivalent emissions are required for these submissions, a default GWP for fully fluorinated GHGs of $10\,000$ (100-yr time horizon) is used due to the lack of a peer-reviewed GWP. Emissions have mainly been reported under the "Fluorinated Gas Production" subpart for 2011–2017 with a maximum of 40 t in 2013 and a subsequent decline to 4.5 t by 2017.

$c$-$C_4F_8O$ has been under discussion in the recent literature foremost as a new Chemical Vapor Deposition (CVD) chamber cleaning agent in the semiconductor industry (Pruette et al., 2000; McCoy, 2000; Oh et al., 2001; Kim et al., 2002, 2004). It was evaluated against the widely used perfluoroethane ($C_2F_6$) and perfluoropropane ($C_3F_8$) in terms of cleaning effectiveness and reduction in greenhouse gas emissions. Its advantages over $NF_3$, another alternative cleaning agent, are stated as lower toxicity and a smaller adjustment to existing chamber cleaning structures using $C_2F_6$. However, its disadvantages are potential byproducts such as tetrafluoromethane ($CF_4$), which is another long-lived potent greenhouse gas (Beu, 2005). $c$-$C_4F_8O$ has also been evaluated as part of a gas mixture to replace $SF_6$ in high-voltage gaseous insulation applications, again driven by the desire to reduce greenhouse gas emissions (Dahl et al., 2014; Chachereau et al., 2016; Kočišek et al., 2018). Another niche application is the use as a radiator gas for Cherenkov detectors in large scale particle acceleration experiments (Artuso et al., 2006; Acconcia et al., 2014).

The above applications have emerged only within the last two decades. Whether $c$-$C_4F_8O$ was used earlier than that is undocumented. Frick and Anderson (1972) patented a method to synthesize $c$-$C_4F_8O$ for potential use as an inert solvent for highly reactive or corrosive halogenated materials in naval applications. However it remains unclear if this led to mass production of $c$-$C_4F_8O$ at that time.

Little is known of $c$-$C_4F_8O$ related to its radiative properties and gas phase loss processes in the atmosphere. A GWP of 8 700 (with no reference to the time horizon) has been reported, which was derived based on structural analogies to octafluoro-cyclobutane ($c$-$C_4F_8$), for which the GWP is known (3M company-internal analysis cited by Pruette et al. (2000)). A Material Safety Data Sheet for PFG-3480 (trade name for $c$-$C_4F_8O$), lists a GWP of 13 900 (100-yr time horizon) and a lifetime of <4 000 years (3M Company, 2007). It is unknown to us how these results were obtained. Information is also lacking on other potential loss processes for $c$-$C_4F_8O$, such as uptake by oceans and land. In addition, no atmospheric measurements, and consequently no atmospheric observation based (top-down) emissions estimates are presently available for $c$-$C_4F_8O$.

The present study aims to improve our knowledge on the chemical and radiative properties of $c$-$C_4F_8O$ relevant to determining its atmospheric lifetime and to provide the first atmospheric measurements from which we derive estimated global emissions to the atmosphere. Measurements were made on atmospheric samples archived in canisters and Antarctic firn, and in modern air from in situ observations. From the derived historical record, emissions are estimated using a 12-box chemical transport model of the atmosphere (Cunnold et al., 1983; Rigby et al., 2013; Vollmer et al., 2016). We also conducted laboratory experiments to determine the infrared and ultraviolet (UV) absorption spectra of $c$-$C_4F_8O$, and the rate coefficient for the $O(^1D)$ + $c$-$C_4F_8O$ reaction to estimate the atmospheric lifetime and GWP of $c$-$C_4F_8O$.

## 2 Methods

### 2.1 $c$-$C_4F_8O$ in air samples

#### 2.1.1 Measurements of $c$-$C_4F_8O$ in archived and ambient in situ air

For the present study, archived and urban ambient air samples were analyzed at the Commonwealth Scientific and Industrial Research Organization (CSIRO) laboratory at Aspendale (Victoria, Australia) using Medusa gas chromatographic (GC) mass spectrometric (MS) techniques (Miller et al., 2008). The archived air samples consisted primarily of the Cape Grim Air Archive (CGAA) samples collected under clean air baseline conditions for archival purposes since 1978 at the Cape Grim Baseline Air Pollution Station (Tasmania, Australia 40.7°S, 144.7°E). These >100 samples were collected into 34 L internally electropolished stainless steel canisters (Essex Industries, USA) using cryogenic techniques (Fraser et al., 1991; Langenfelds et al., 1996, 2014; Fraser et al., 2016). The CGAA record was complemented with a few samples collected in the Northern Hemisphere (33°N–53°N) mostly using oil-free diving compressors.

Two firn air samples were also analyzed, which were collected at the Aurora Basin North site in Antarctica (71.1°S, 111.4°E). The site is located 550 km inland from Australia's Casey station, at 2710 masl and has a low mean annual air temperature of −44°C. Samples were collected in December 2013; those for the halocarbon measurements were collected into internally electropolished stainless steel containers using a 2-stage teflon-coated viton diaphragm pump. Only two samples were available for the present study as other samples from this site were used for a different halocarbon study.

In situ measurements of $c$-$C_4F_8O$ at Aspendale (38.0°S, 145.1°E) were started in February 2017. These samples are collected from the rooftop at CSIRO (at 11 m height from the ground) through a 3/8" OD Synflex 1300 tube (Saint-Gobain, France) using

a continuous flow air sampling module (Miller et al., 2008) with a diaphragm sampling pump fitted with stainless steel heads and a neoprene membrane (KNF Neuberger, Germany).

All archived air samples were analyzed on the Medusa-GCMS "Medusa-9" in December 2016 along with a suite of other trace gases. The instrument is based on the original design of the Medusa-GCMS used in the Advanced Global Atmospheric Gases Experiment (AGAGE) network (Miller et al., 2008; Prinn et al., 2018) but fitted with different chromatography columns (Vollmer et al., 2018). A GS-GasPro main capillary column (0.32 mm ID × 60 m, Agilent Technologies) was used for the main separation and a column of the same type (5 m) was fitted as a precolumn, allowing for a backflushing of late eluting compounds. In this GCMS setup (Agilent 6890 GC, 5975 MS) $c$-$C_4F_8O$ was identified using a multi-component diluted mixture of known composition with the MS in scan and selected ion modes. The choice for the two fragments used in the analysis of our air samples was based on the mass spectrum, which we measured for $c$-$C_4F_8O$, to the best of our knowledge the first one published for this compound (see Supplement).

Analytes from the samples were cryogenically preconcentrated on a first microtrap of the GCMS and subsequently transferred to a second microtrap, both filled with HayeSepD and held at $\sim -155\,°C$. During this process, water vapor was largely removed using nafion dryers; nitrogen, oxygen, and a large fraction of noble gases were removed due to their trap breakthroughs, and carbon dioxide was removed using a molecular sieve (4A) packed column between the traps. To enhance the signal size of the measured compounds, 3 L sample sizes were used for each measurement (compared to normally 2 L) and the MS electron multiplier voltage was increased by 50 V compared to what was given by the autotune algorithm. Analysis of a single sample lasted 65 min. Archived air sample measurements were bracketed by measurements of a standard (E-146S) to track and correct for MS sensitivity changes. This standard was air compressed into a 34 L tank at the remote Rigi-Seebodenalp station (Switzerland) using an oil-free compressor, and was additionally spiked with small amounts of $c$-$C_4F_8O$ and other compounds, to enhance the GCMS peak size and signal-to-noise ratio. In general, three measurements of each archived air sample were made. For some, no standard measurement was made between the second and third sample to assess potential memory effects of the system. For $c$-$C_4F_8O$, no memory effect and no signal in the blank runs could be detected. Detection limits are estimated at 5 ppq (parts per quadrillion, femtomol mol$^{-1}$). Mean precisions ($2\sigma$) for the measurements of the archived air samples ranged 3–4 ppq (20–5 %) for the low ($\sim$15 ppq) to high ($\sim$70 ppq) mole fractions, respectively. Based on two different types of experiments, a linear system response for the relevant mole fraction range was found (see Supplement). In situ urban air measurements at Aspendale are based on 2 L samples and without alteration of the MS electron multiplier voltage. Consequently the precisions are slightly poorer for these measurements. These air precisions were estimated at $\sim$12 ppq ($\sim$17 %, $2\sigma$) under the assumption that $c$-$C_4F_8O$ remains constant in the air measured in situ at Aspendale on a daily basis.

### 2.1.2 Absolute calibration and uncertainty estimates for air measurements

A primary calibration scale was prepared based on a commercially obtained multi-component mixture in dry synthetic air (Carbagas, Switzerland, HCP-04Carba), with a mole fraction of $c$-$C_4F_8O$ at 10 ppm (parts-per-million, $\mu$mol mol$^{-1}$). This mixture was diluted manometrically and using a bootstrap technique, resulting in a primary calibration standard (EP-001) with $c$-$C_4F_8O$ at 1.81 ppt (parts-per-trillion, picomol mol$^{-1}$). Three secondary standards were additionally prepared from ambient

air compressed into cylinders (Essex Industries, USA) and spiked with small quantities of $c$-$C_4F_8O$ resulting in mole fractions of ~0.5 ppt. These secondary standards were the base for propagating the calibration scale to other calibration standards, in particular that used for the Cape Grim Air Archive measurements (E-146S). They define the Empa-2013 calibration scale for $c$-$C_4F_8O$ on which our results are reported. The systematic uncertainty of the preparation of this primary calibration scale (including its propagation to the working standards), which defines its accuracy, is estimated at 15 % ($2\sigma$). Details of the dilution technique and the primary calibration scale are provided by Vollmer et al. (2015).

## 2.2 Models and inversion

### 2.2.1 Firn model

We use a numerical firn air model (Trudinger et al., 1997, 2013) to quantify the movement of $c$-$C_4F_8O$ in firn air in order to determine the time period for which $c$-$C_4F_8O$ in the firn samples is representative of the atmosphere. Vertical diffusion in the firn and other physical processes cause a tracer in a firn air sample to correspond to an age spectrum relative to the atmosphere, rather than a discrete age. Green's functions from the firn model represent the age spectrum of a tracer in each firn sample, and are used in this work to relate the measured mole fractions of $c$-$C_4F_8O$ in firn to the time-range of the corresponding atmospheric mole fractions.

For Aurora Basin North, the firn model uses an accumulation rate of 97 kg m$^{-2}$ yr$^{-1}$, a temperature of $-44°$C, and pressure of 695 hPa. The density profile used was based on a spline fit to density measurements. Diffusion parameters in the firn model are calibrated for Aurora Basin North using 12 tracers at between 5 and 11 depths each throughout the firn. Only molecular diffusion was used for the Aurora Basin North firn model calculations; eddy diffusivity is sometimes used in the CSIRO firn model in the deep firn but was not used here as the parameters were not well constrained by the available measurements. The diffusion coefficient used in this work for $c$-$C_4F_8O$ in air, relative to $CO_2$ in air, (for a temperature of 253 K) is 0.460. This value was determined using Equation 4 from Fuller et al. (1966) with Le Bas volume increments (e.g. Table 1.3.1, Mackay et al. (2006)) and a multiplier for the Le Bas increments of 0.97 (this value minimizes the difference of calculated relative diffusion coefficients of a number of compounds from values measured by Matsunaga et al. (1993, 2002, 2005)).

### 2.2.2 12-box atmospheric model

We use the AGAGE 12-box atmospheric model (Rigby et al., 2013) to relate the atmospheric mole fractions to surface emissions. Briefly, in this model, the atmosphere is divided into four zonal bands, separated at the equator and at the 30° latitudes, thereby creating boxes of similar air masses. There are also vertical separations, at altitudes represented by 500 hPa and 200 hPa, resulting in the overall 12 boxes. Model transport parameters and stratospheric photolytic loss vary seasonally and repeat interannually (Rigby et al., 2013). We anticipate that variations in emissions dominate atmospheric trends, particularly over the longer (multi-annual) timescales that are our primary focus, so inter-annual variation in transport is not expected to be important here. Loss processes other than those in the atmosphere, such as uptake by land and oceans, and potential natural sources, are not included in the model. Green's functions derived from the 12-box atmospheric model relate atmospheric mole

fraction in the high-latitude Northern and Southern Hemispheres to annual global emissions in preceding years, and are used in the inversion (Trudinger et al., 2016).

### 2.2.3 Global inversions

To estimate global emissions to the atmosphere from the mole fraction measurements, we employ an inverse calculation (in-
version InvE2 from Trudinger et al. (2016), and termed "CSIRO" inversion in Vollmer et al. (2016, 2018)) that was developed
to focus on sparse observations from air archives, and firn air and ice core samples that are associated with age spectra. The
inversion combines Green's functions from both the firn model and AGAGE 12-box atmospheric model described above to
relate firn and tropospheric mole fraction to $c$-$C_4F_8O$ surface emissions. The Green's functions from the 12-box model were
calculated using a constant distribution of emissions into the four zonal boxes at the surface, and for this we used the rela-
tive contributions 0.675, 0.325, 0.0, and 0.0, in the northern-most to southern-most zonal bands. Results are fairly insensitive
to emissions distributions that have most emissions in the Northern Hemisphere (see Supplement). The characteristics of
sparse atmospheric, firn and ice core data necessitate the use of constraints on the inversion to avoid unrealistic oscillations
in the reconstructed mole fractions or negative values of mole fraction or emissions. The inversion uses non-negativity con-
straints and favors relatively small changes in annual emissions between adjacent years over large, unrealistic fluctuations.
A prior emissions history is needed as a starting point for the inversion; then a non-linear constrained optimization method
(*constrained_min* routine in IDL (Harris Geospatial Solutions, Inc., Broomfield, Colorado) is used to find the solution that
minimizes a cost function consisting of the model-data mismatch weighted by the observation uncertainties, plus the sum of
the year-to-year changes in emissions (Trudinger et al., 2016). Given the lack of industry-based bottom-up emission estimates
for $c$-$C_4F_8O$, we use emissions derived from observations of perfluorooctane, which was found present for many decades and
at low abundances in the global atmosphere (Ivy et al., 2012). Because the prior is not based on information on $c$-$C_4F_8O$, we
do not include the prior in the cost function. The emissions derived from the inversion are rather insensitive to the choice of
the prior (see Supplement), because the prior is used here as a starting point for the inversion only, and not as a constraint. Our
observations used in the inversion are the firn measurements and annual values of mole fraction from a smoothing spline fit
(50 % attenuation at 10 years) to measurements of the CGAA and in situ measurements at Aspendale. Northern Hemisphere
measurements were compared with the reconstructed mole fractions for that hemisphere, but were not used in the inversion.
Uncertainties in the emissions are estimated using a bootstrap method that incorporates temporally-correlated uncertainties in
the annual values derived from the atmospheric data (see Supplement), uncertainty in firn measurements, uncertainty of $\pm15$ %
in the calibration scale, and uncertainties in the firn model parameters through the use of an ensemble of firn Green's functions.

### 2.3 Laboratory studies

Laboratory studies to measure the infrared and UV spectra of $c$-$C_4F_8O$ and the rate coefficient for the $O(^1D)$ + $c$-$C_4F_8O$ reac-
tion were conducted at the Chemical Sciences Division Laboratories at the National Oceanic and Atmospheric Administration
(NOAA), Boulder, Colorado, USA. The apparatus and methods used in this work are described separately below.

### 2.3.1 Absorption spectra

Absorption spectra, $A(v)$ (base e), or integrated band strengths, were quantified using Beer's law

$$A(\lambda) = -ln\left(\frac{I(\lambda)}{I_0(\lambda)}\right) = \sigma(\lambda) \times L \times [c\text{-}C_4F_8O] \tag{I}$$

where $A$ is the measured absorbance at wavelength $\lambda$, $I(\lambda)$ and $I_0(\lambda)$ are the measured light intensities with and without the sample present in the absorption cell, respectively, $L$ is the optical absorption path length, $\sigma(\lambda)$ is the infrared or UV cross section of $c\text{-}C_4F_8O$, and $[c\text{-}C_4F_8O]$ is the concentration of $c\text{-}C_4F_8O$. In total, 11 independent absorption spectrum measurements were used in the linear least-squares fit. The $c\text{-}C_4F_8O$ concentration was determined using the ideal gas law and absolute pressure measurements of either the pure compound or of a dilute mixture of the compound in a helium (He) bath gas.

The $c\text{-}C_4F_8O$ sample was obtained from SynQuest Laboratories Inc. (Alachua, Florida, USA, 99 % purity). For the experiments described below, $c\text{-}C_4F_8O$ was introduced into the absorption cells as a pure sample or in a dilute mixture prepared off-line. The dilute mixtures of $c\text{-}C_4F_8O$ in a He (UHP, 99.999 %) bath gas were prepared manometrically in a 12 L Pyrex bulb with an estimated accuracy of $\sim$1 %, as derived from the accuracy of the pressure measurements. The pressure measurement uncertainty includes the uncertainty in the sample pressure, the total mixture pressure, and the linearity of the pressure gauge (estimated to be 0.2 %). Pressures were measured with 100 Torr and 1 000 Torr (130 and 1 300 hPa, respectively) capacitance manometers. Quoted uncertainties are $2\sigma$.

Infrared absorption spectra were measured at 296 K using Fourier transform infraRed (FTIR) spectroscopy over the 500–4 000 cm$^{-1}$ spectral region at 1 cm$^{-1}$ resolution with Boxcar apodization. The apparatus has been used extensively in previous studies (Bernard et al., 2017, 2018). The FTIR was coupled to a 15 cm path length single pass absorption cell with potassium bromide (KBr) windows. A liquid-nitrogen cooled HgCdTe/B semiconductor detector was used. Infrared spectra were recorded in 100 or 500 co-added scans. Absorption spectra were recorded under static conditions using a dilute mixture of $c\text{-}C_4F_8O$ in He with a 0.00180 mixing ratio. The $c\text{-}C_4F_8O$ concentration used in the absorption measurements was in the range $1.75 \times 10^{15}$ to $2.34 \times 10^{16}$ molecule cm$^{-3}$. Integrated band strengths (IBS) were obtained from the measurement of 11 individual IR spectra.

The UV absorption spectrum of $c\text{-}C_4F_8O$ was measured at 296 K using a 0.5 m spectrometer (1 nm resolution) equipped with a charge-coupled device (CCD) detector. The collimated output of a 30 W deuterium lamp passed through a 100 cm long and 2.5 cm diameter Pyrex absorption cell with quartz windows. Spectral measurements were made over the wavelength region 200–350 nm. The wavelength scale of the spectrometer was calibrated using the emission lines from a low-pressure Hg pen-ray lamp. $c\text{-}C_4F_8O$ was added to the absorption cell in pure form from the original sample. Measurements were performed over a range of $c\text{-}C_4F_8O$ concentrations from $2.51 \times 10^{18}$ to $2.16 \times 10^{19}$ molecule cm$^{-3}$. Eleven independent UV absorption spectrum measurements were used in the final linear least-squares fit.

### 2.3.2 O($^1$D) reaction rate coefficient

The reactive rate coefficient, $k_1$, for the reaction

$$O(^1D) + c\text{-}C_4F_8O \quad \rightarrow Products \tag{1a}$$

$$O(^1D) + c\text{-}C_4F_8O \quad \rightarrow c\text{-}C_4F_8O + O(^3P) \tag{1b}$$

i.e., the channel resulting in the loss of $c$-C$_4$F$_8$O, was measured at 294 K using a relative method (e.g. Baasandorj et al., 2013). The loss of $c$-C$_4$F$_8$O was measured relative to the loss of the reference compound CHF$_3$ during the same experiment:

$$O(^1D) + CHF_3 \quad \rightarrow Products \tag{2a}$$

$$O(^1D) + CHF_3 \quad \rightarrow CHF_3 + O(^3P) \tag{2b}$$

The recommended total rate coefficient for reaction 2, $k_2$ is $(9.60\pm0.5)\times10^{-12}$ cm$^3$ molecule$^{-1}$ s$^{-1}$ and the recommended
reactive channel branching ratio, $k_{2a}/k_2$ is 0.25, i.e., $k_{2a} = 2.4\times10^{-12}$ cm$^3$ molecule$^{-1}$ s$^{-1}$ (Burkholder et al., 2015).

Provided $c$-C$_4$F$_8$O and the reference compound are removed solely by reaction with O($^1$D), the rate coefficient for reaction 1a is related to the reference compound rate coefficient by the equation

$$ln\left(\frac{[c\text{-}C_4F_8O]_{t_0}}{[c\text{-}C_4F_8O]_t}\right) = \frac{k_{1a}}{k_{2a}}\left[ln\left(\frac{[CHF_3]_{t_0}}{[CHF_3]_t}\right)\right] \tag{II}$$

where $[c\text{-}C_4F_8O]_{t_0}$, $[CHF_3]_{t_0}$, $[c\text{-}C_4F_8O]_t$, and $[CHF_3]_t$ are the concentrations of $c$-C$_4$F$_8$O and CHF$_3$ at times zero ($t_0$) and
$t$, respectively. The $k_{1a}$ and $k_{2a}$ are the reactive rate coefficients for the reaction of O($^1$D) with $c$-C$_4$F$_8$O (1a) and CHF$_3$ (2a), respectively.

The Pyrex reactor, which was 100 cm long and with a 2.2 cm internal diameter, was coupled with a Teflon circulating pump to an absorption cell where the losses of $c$-C$_4$F$_8$O and CHF$_3$ were measured using FTIR spectroscopy. The FTIR absorption cell was equipped a multi-pass cell (485 cm path length) with KBr windows. Spectra were recorded in 100 co-adds at a spectral
resolution of 1 cm$^{-1}$.

O($^1$D) was produced by KrF (248 nm) excimer pulsed laser photolysis of ozone:

$$O_3 + h\nu\,(248\,nm) \rightarrow O(^1D) + O_2(^1\Delta) \tag{3a}$$

$$\rightarrow O(^3P) + O_2(^3\Sigma) \tag{3b}$$

The yield of the O($^1$D) channel is 0.9 (Burkholder et al., 2015). After thoroughly mixing the gas mixture in the system,
a time zero infrared spectrum was recorded. Ozone was then added slowly to the reactor with the photolysis laser and gas circulation on. The photolysis laser fluence was in the range $\sim$2–7.4 mJ cm$^{-2}$ pulse$^{-1}$. The laser was operated at 10 or 20 Hz. The total pressure in the cell increased during an experiment by $\sim$300 Torr, mostly due to the addition of He carrier gas used to flush ozone into the reactor. Infrared spectra were recorded at regular intervals with approximately 10 spectra recorded over the course of an experiment. Experiments performed separately demonstrated that there was no significant loss of $c$-C$_4$F$_8$O or
CHF$_3$ under identical conditions in the absence of photolysis. The initial $c$-C$_4$F$_8$O and CHF$_3$ concentrations were in the range 6.4–6.8$\times10^{14}$ molecule cm$^{-3}$ and 4.5–5.0$\times10^{14}$ molecule cm$^{-3}$, respectively.

## 3 Results and discussion

### 3.1 Infrared spectrum

The infrared absorption spectrum of $c$-$C_4F_8O$ obtained in this study is shown in Fig. 1. Over the range of $c$-$C_4F_8O$ concentrations used, the spectra obeyed Beer's law with high precision ($\sim$0.2 %). Spectra recorded at different total pressures had
identical band shapes, i.e., the spectrum was independent of the total pressure (He bath gas) over the range of 30–400 Torr.

The integrated band strength (IBS) over the spectral region 500–1 500 $cm^{-1}$ was determined to be $(3.21 \pm 0.01) \times 10^{-16}$ $cm^2$ molecule$^{-1}$ cm$^{-1}$, where the quoted uncertainty is the precision of the linear least-squares fit of the data to Beer's law (Equation I). The absolute uncertainty in the $c$-$C_4F_8O$ spectrum includes estimated uncertainties in the optical path length ($\pm$0.5 %), measured absorbance ($\pm$0.005), temperature ($\pm$1 K), and pressure ($\pm$0.2 %). The absolute uncertainty in the total
integrated band strength is estimated to be 3 %.

### 3.2 UV absorption

UV absorption of $c$-$C_4F_8O$ was observed between 200 nm and 225 nm, a range that is most critical for calculations of the $c$-$C_4F_8O$ atmospheric photolysis rates. The spectrum is continuous, with cross sections decreasing monotonically with increasing wavelength. The cross section measurements obeyed Beer's law with values of $(9.2 \pm 3.9) \times 10^{-24}$ and $(4.4 \pm 2.3) \times 10^{-24}$
$cm^2$ molecule$^{-1}$ cm$^{-1}$ at 200 and 225 nm, respectively. An example of the measured UV spectrum of the $c$-$C_4F_8O$ sample is shown in the Supplement. Overall, the cross sections of $c$-$C_4F_8O$ were very low, and therefore, the measurements are susceptible to interference from even minor sample impurities. Therefore, we choose to assign a conservative UV cross section of $<2 \times 10^{-23}$ $cm^2$ molecule$^{-1}$ cm$^{-1}$ over the 200–225 nm range.

### 3.3 O($^1$D) reaction

We found the reactivity of $c$-$C_4F_8O$ with O($^1$D) to be low, which makes the determination of an accurate rate coefficient more challenging. The relative rate data are shown in Fig. 3 and tabulated in the Supplement. The precision of the three independent measurements is high with a fit precision of a few percent. However the agreement between the independent measurements is relatively poor. The low conversion of $c$-$C_4F_8O$, $<2$ %, and the precision of the infrared spectral subtractions are the primary sources of uncertainty in the measurements. The low conversion of $c$-$C_4F_8O$ and $CHF_3$ achieved in these
experiments was primarily due to the build up of $O_2$ associated with the addition of ozone to the reactor, making the loss of O($^1$D) by reaction with $O_2$ more significant than its reaction with $c$-$C_4F_8O$ and $CHF_3$. The absorption bands used in this study were 1120–1000 and 1180–1120 $cm^{-1}$ for $c$-$C_4F_8O$ and $CHF_3$, respectively. The spectral subtraction uncertainty is illustrated by the error bars included in Fig. 3. The average of individual measurements yields a rate coefficient ratio of $0.21\pm0.07$. However, we recommend a conservative upper-limit of 0.5. Using the recommended O($^1$D) + $CHF_3$ reactive rate coefficient,
$(2.41\pm0.12)\times10^{-12}$ $cm^3$ molecule$^{-1}$ s$^{-1}$ (Burkholder et al., 2015) yields a $c$-$C_4F_8O$ reactive rate coefficient of $<1.2\times10^{-12}$

cm$^3$ molecule$^{-1}$ s$^{-1}$. Given the small O($^1$D) rate coefficient of $c$-C$_4$F$_8$O, a refinement of the measured O($^1$D) reactive rate coefficient will have a negligible impact to the total atmospheric loss of $c$-C$_4$F$_8$O.

## 3.4 Atmospheric lifetime

The global annually averaged atmospheric lifetime ($\tau$) of $c$-C$_4$F$_8$O, is defined with respect to the individual partial lifetimes by the relationship:

$$\frac{1}{\tau} = \frac{1}{\tau_{O(^1D)}} + \frac{1}{\tau_{h\nu}} + \frac{1}{\tau_{Lyman-\alpha}} + \frac{1}{\tau_{OH}} \tag{4}$$

where the individual global loss processes are combined to derive the overall global lifetime. In the present analysis, only O($^1$D), UV photolysis, and Lyman-$\alpha$ terms are considered. This study has focused primarily on the atmospheric loss processes of $c$-C$_4$F$_8$O, i.e., potential deposition or heterogeneous loss processes of $c$-C$_4$F$_8$O were beyond the scope of this study. Deposition or heterogeneous loss processes, if significant, would lead to a shorter global lifetime for $c$-C$_4$F$_8$O. The loss of $c$-C$_4$F$_8$O via reaction with the OH radical is assumed to make a negligible contribution to the global lifetime in our analysis. The OH rate coefficient would need to be $\sim$1$\times$10$^{-17}$ cm$^3$ molecule$^{-1}$ s$^{-1}$ (equivalent to a 3 000 year lifetime) to make a significant global lifetime contribution. Such a low rate coefficient represents a significant challenge to current rate coefficient measurement methods. Additionally, an expected OH low-reactivity of $c$-C$_4$F$_8$O is supported by its low O($^1$D) reactivity measured in this work. Consequently, we ignore the last term in Equation 4. Additional laboratory studies, that are beyond the scope of the present work, would be needed to quantify the OH reaction.

The laboratory results for the O($^1$D) reaction and UV photolysis obtained in this study are combined with an estimated Lyman-$\alpha$ lifetime to derive the lifetime utilizing the 2-D atmospheric model calculation parametrizations reported by Bernard et al. (2019). The O($^1$D) reaction represents a stratospheric loss process for $c$-C$_4$F$_8$O with a partial lifetime, $\tau_{O(^1D)}$, estimated to be >30 000 years. This extremely long lifetime is a result of the low $c$-C$_4$F$_8$O reactivity combined with the turn-over time of the stratosphere. The UV photolysis lifetime, $\tau_{h\nu}$ is derived from the $c$-C$_4$F$_8$O UV cross section upper-limit of 10$^{-23}$ cm$^2$ molecule$^{-1}$ and an assumed quantum yield of unity for the 200–225 nm region, and is >15 000 years. Combining the estimated O($^1$D) and UV photolysis lifetimes yields an estimated global lifetime for $c$-C$_4$F$_8$O >7 500 years.

Given the long atmospheric lifetime of $c$-C$_4$F$_8$O in the troposphere and stratosphere, upper atmospheric loss processes may contribute to the global atmospheric lifetime. Here, we consider loss due to Lyman-$\alpha$ photolysis, although other loss processes are possible. To date, the vacuum UV (VUV) absorption spectrum of $c$-C$_4$F$_8$O, which includes the Lyman-$\alpha$ absorption (121.567 nm), has not been reported. The scope of the present study did not include a measurement of the Lyman-$\alpha$ cross section. It is however reasonable to assume a Lyman-$\alpha$ cross section of $\sim$10$^{-17}$ cm$^2$ molecule$^{-1}$ for $c$-C$_4$F$_8$O which is in the range of values for highly fluorinated compounds of (0.035–10)$\times$10$^{-17}$ cm$^2$ molecule$^{-1}$ (SPARC, 2013). Therefore, in the absence of experimental data, we consider the Lyman-$\alpha$ cross section 1$\times$10$^{-17}$ cm$^2$ molecule$^{-1}$, used in our lifetime analysis, as a reasonable estimate. Note that a smaller (larger) Lyman-$\alpha$ cross section would lead to a longer (shorter) photolysis lifetime, although the lifetime dependence on the cross section value is not linear due to the lifetime dependence on the transport time to the mesosphere. On the basis of these assumptions, Lyman-$\alpha$ photolysis in the lower mesosphere could be the dominant atmo-

spheric loss process for $c$-C$_4$F$_8$O. Including this Lyman-$\alpha$ photolysis lifetime yields a $c$-C$_4$F$_8$O globally averaged atmospheric lifetime of >3 000 years.

## 3.5 Global warming potential (GWP)

$c$-C$_4$F$_8$O has strong vibrational absorption bands within the Earth's atmospheric infrared transmission window (Hodnebrog et al.

(2013), Fig. 4). We determine a radiative efficiency for $c$-C$_4$F$_8$O of 0.430 W m$^{-2}$ ppb$^{-1}$ using the parameterization for atmospherically well-mixed compounds given in Hodnebrog et al. (2013). $c$-C$_4$F$_8$O is therefore a potent greenhouse gas with a radiative efficiency greater than those of HFCs and chlorofluorocarbons (CFCs), which are typically less than 0.3 W m$^{-2}$ ppb$^{-1}$ (Myhre et al., 2013).

The GWP of $c$-C$_4$F$_8$O was calculated using the global atmospheric lifetime lower-limit of 3 000 years and the radiative

efficiency determined in this work:

$$\text{GWP(T)} = \frac{\text{RE}_\tau \left(1 - \exp(-\text{T}/\tau)\right)}{\text{M}_{c\text{-C}_4\text{F}_8\text{O}} \int \text{RF}_{\text{CO}_2}(\text{T})} \tag{5}$$

where RE is the radiative efficiency, T is the time horizon (in years), $\text{M}_{c\text{-C}_4\text{F}_8\text{O}}$ is the molar weight of $c$-C$_4$F$_8$O, and $\text{RF}_{\text{CO}_2}$ is the radiative forcing of CO$_2$. The GWPs are 8 975, 12 000, and 16 000 for the 20, 100, and 500-year time-horizons. Therefore, $c$-C$_4$F$_8$O is a potent radiative forcing agent due to the combination of its high radiative efficiency and long atmospheric lifetime.

The GWPs for $c$-C$_4$F$_8$O are comparable to the values for long-lived perfluorocarbons (PFCs) that have GWP$_{100}$ values in the range 6 300–11 100 (Harris and Wuebbles, 2014).

## 3.6 Atmospheric observations and emissions of $c$-C$_4$F$_8$O

We observe a general increase of $c$-C$_4$F$_8$O in the atmosphere over the sample period starting in 1978 (Fig. 2). $c$-C$_4$F$_8$O was detectable in all samples but its abundance was low in the early record (<20 ppq) until about 1998, when its growth rate

increased strongly. Its abundance increased rather steadily for more than a decade but has subsequently leveled at 73–75 ppq in 2015–2018. The growth rate was at a maximum of 4.0 ppq yr$^{-1}$ in 2004 and declined from that to <0.25 ppq yr$^{-1}$ in 2017 and 2018 as a consequence of the relatively constant abundance in the last few years.

The $c$-C$_4$F$_8$O measurements in the Southern Hemisphere provide a strong constraint on the trend in both hemispheres due to the very long lifetime of $c$-C$_4$F$_8$O in the atmosphere, the relatively rapid mixing of the atmosphere, and the expectation

that most $c$-C$_4$F$_8$O emissions are in the Northern Hemisphere. Most anthropogenic gases are released predominantly in the Northern Hemisphere, including gases released by the semiconductor industry. The assumption of mainly Northern Hemisphere emissions for $c$-C$_4$F$_8$O leads to higher mole fraction values in the Northern Hemisphere than in the Southern Hemisphere, and this is confirmed by comparison of the modeled Northern Hemisphere history with the few Northern Hemisphere samples that we do have. In the Supplement, we test the sensitivity of inferred global emissions and mole fraction in both hemispheres to

the assumed spatial distribution of emissions.

The two firn air samples fit well into the CGAA record with the older sample at slightly lower mole fraction than the oldest CGAA samples. Due to the very long lifetime, this suggests that $c$-C$_4$F$_8$O was below 10 ppq in the Southern Hemisphere

before 1978, and could only have been steady or increasing. However it is impossible to further pin down the first appearance of this compound in the atmosphere and the exact course of the abundance until ~1980 because our knowledge of $c$-$C_4F_8O$ prior to the CGAA is based on only one firn sample measurement with air spanning several decades (see calculated Green's functions in the Supplement). Also, potential small contamination during firn air sampling by modern air or sampling devices

cannot be fully excluded, and the measurement of the older firn air sample is close to the instrument's detection limit. Given these limitations, we are not able to draw any conclusions on any potential naturally-occurring $c$-$C_4F_8O$. Nevertheless, the two firn air sample measurements allow us to draw conclusions on storage stability of $c$-$C_4F_8O$ in canisters. Given that the storage time of the two firn air samples in the canisters is much shorter than those of the older CGAA samples, the good agreement of the firn air results with those of the CGAA is supportive of storage stability of $c$-$C_4F_8O$ in the CGAA tanks and confirms that

the observed multidecadal record is not a simple artifact of degradation of $c$-$C_4F_8O$ in canisters over time.

In situ measurements at Aspendale, which are available on a regular measurement basis since February 2017, show a constant abundance of $c$-$C_4F_8O$ at ~74 ppq. This lack of growth is an indication of currently very small, if at all any, emissions of this compound. Also, pollution events are absent from this urban in situ record within the precision of these measurements, suggesting that $c$-$C_4F_8O$ is not emitted within the airmass footprints of the site.

Emissions derived from the atmospheric observations were low during the first part of our record (Fig. 2). Until 1980, when the global mean abundance of $c$-$C_4F_8O$ was 10 ppq, cumulative emissions had reached 0.38 kt. For the time after ~1980, when observations became more frequent, emissions were 0.02–0.03 kt yr$^{-1}$ for about a decade. From the mid 1990s, emissions increased strongly to a maximum of 0.15 ($\pm$0.04, 2 $\sigma$) kt yr$^{-1}$ in 2004. Surprisingly, emissions have declined since, to <0.015 kt yr$^{-1}$ by 2017 and 2018. This rapid decline is suggestive of a switch to alternative compounds in large scale industrial

applications such as in the semiconductor industry, or of a better containment in these applications or where $c$-$C_4F_8O$ might be emitted as byproduct.

For the USA, reported bottom-up emissions of $c$-$C_4F_8O$ under the GHGRP subpart "Fluorinated Gas Production" for 2011–2017 are surprisingly high (about half) compared to our top-down emissions estimates (Fig. 2c). It therefore appears unlikely that these bottom-up emissions derive entirely from fugitive emissions during $c$-$C_4F_8O$ production, but they could, at least in

partial, derive from emissions of $c$-$C_4F_8O$ as a byproduct during the production of other fluorochemicals. Anyhow, they reveal a surprisingly similar relative decline compared to our top-down estimates.

Cumulative emissions until 2018 amount to 2.8 (2.4–3.3) kt. If scaled with the GWP on a 100-yr time horizon, as derived below, they correspond to 34 Mt $CO_2$-equivalents. Despite the high GWP, these emissions are small compared to the major greenhouse gases but of similar magnitude to some of the other minor greenhouse gases such as minor perfluorocarbons and

fluorinated inhalation anesthetics (Ivy et al., 2012; Vollmer et al., 2015). Whether these cumulative emissions remain at low levels will depend on potential future choices for $c$-$C_4F_8O$ in large scale applications.

## 4 Conclusions

We provide first laboratory experiments of atmospheric loss processes and first atmospheric observations of $c$-$C_4F_8O$. We measured infrared and UV absorption spectra of $c$-$C_4F_8O$, and the rate coefficient for the $O(^1D)$ + $c$-$C_4F_8O$ reaction. These experimental results suggest that $c$-$C_4F_8O$ is an atmospherically persistent trace gas with an atmospheric lifetime of >3 000 years. In addition, its strong absorption in the "atmospheric window" results in a very high radiative efficiency, and when combined with the long atmospheric lifetime, yields a high GWP of 12 000 (100-year time horizon), which is exceeded by only a few other greenhouse gases. We show an increase of $c$-$C_4F_8O$ in the atmosphere to present mole fractions of $\sim$75 ppq. Emissions, which were derived from these observations, have strongly declined after a peak in 2004. The reasons for this recent decline, and whether this is only a temporary feature, remain speculative. We hypothesize that the emissions decline could be, at least in partial, a result of the industry's choice for alternative substances for chemical vapor chamber cleaning, which we assumed to have been its primary use in the last two decades. However, even if emissions were completely halted, it will, under the assumption of insignificant non-atmospheric sinks, take thousands of years for $c$-$C_4F_8O$ to be removed from the atmosphere.

Data availability: Data used in this study are available from the Supplement. Mention of trade names or commercial products does not constitute an endorsement or recommendation for use by the authors' affiliated organizations.

Author contributions: MKV, FB, PBK, LPS, SR, and JBB were responsible for the overall project design. RLL, LPS, PJF, BM, and PBK, and DME and MAJC provided the Cape Grim Air Archive and firn air samples, respectively, MKV, BM, and LPS provided the air sample measurements, FB and JBB provided the laboratory measurements and analysis, CMT provided the modeling part for the atmospheric observations. The manuscript was written by MKV and FB with contributions from all co-authors.

Competing interests: The authors declare that they have no conflict of interests.

*Acknowledgements.* We acknowledge the station personnel at Cape Grim for flask sampling for the Cape Grim Air Archive (CGAA) project, which is jointly operated by CSIRO and the Australian government Bureau of Meteorology. CSIRO research at Cape Grim is supported by CSIRO, Bureau of Meteorology, the Australian government Department of Environment and Energy, and Refrigerant Reclaim Australia. We gratefully acknowledge the field team and the national research programs that supported the Aurora Basin North drilling project. M. K. V. acknowledges support from the Swiss Federal Office for the Environment (FOEN) within the Swiss National Programs HALCLIM and CLIMGAS-CH, and 2016 grants from Empa and the Swiss National Science Foundation (SNSF) for technical development and archived air and firn air measurements at CSIRO Aspendale. The work at NOAA was supported in part by the NOAA Climate Office's Atmospheric Chemistry, Carbon Cycle, and Climate Program. General instrument support is provided by members of the Advanced Global Atmospheric Gases Experiment (AGAGE). We acknowledge the contribution by Peter Salameh, who developed and supports GCWerks, a software control system used in the Medusa-GCMS measurements of air samples described here. Deborah Ottinger is acknowledged for her clarifications related to the USA GHG reporting regulations, and Matt Rigby for his contributions to the AGAGE 12-box model. We acknowledge two anonymous reviewers for their detailed and thorough comments on the manuscript.

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

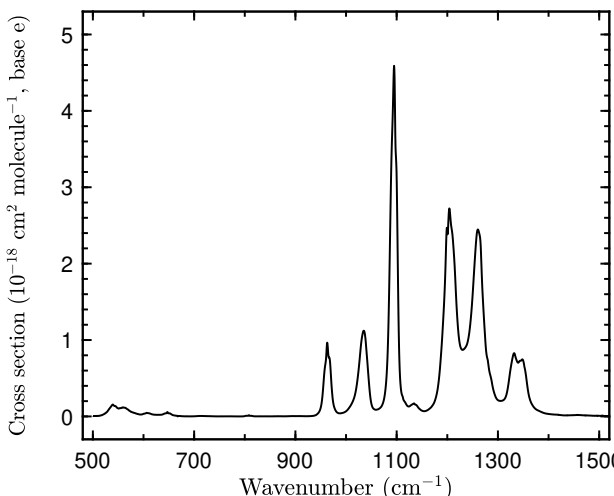

**Figure 1.** Infrared absorption spectrum of octafluorooxolane ($c$-$C_4F_8O$) measured in this work at 296 K at 1 cm$^{-1}$ resolution.

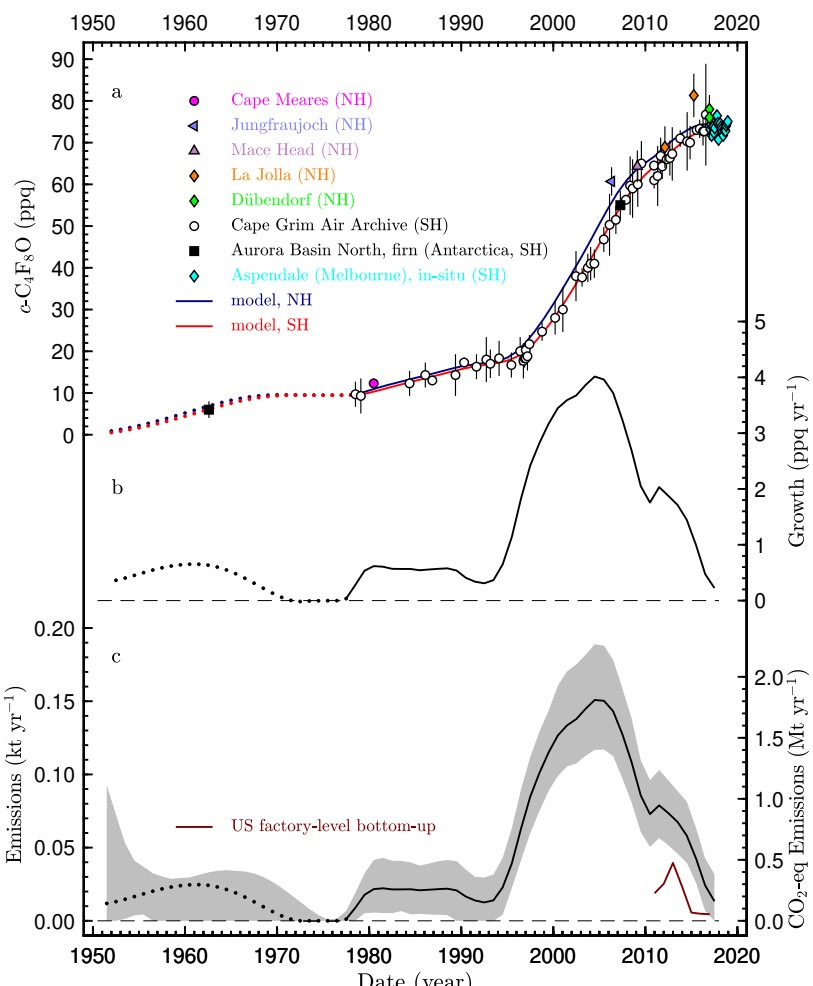

**Figure 2.** Atmospheric observations (a), growth rates (b), and emissions (c) of octafluorooxolane (octafluorotetrahydrofuran, $c$-$C_4F_8O$). Abundances are given as dry air mole fractions in ppq (femtomol $mol^{-1}$) on the Empa-2013 primary calibration scale. Vertical bars denote the measurement precision ($2\sigma$) for the flask samples. Emission uncertainties are $2\sigma$. Samples were collected in the Northern Hemisphere at Cape Meares (Oregon, 45.5°N, 124.0°W), Jungfraujoch (Switzerland, 46.5°N, 8.0°E), Mace Head (Ireland, 53.3°N, 9.9°W), La Jolla (California, 32.9°N, 117.3°W), and Dübendorf (Switzerland, 47.4°N, 8.6°E). Southern Hemisphere samples were mainly from the Cape Grim Air Archive collected at Cape Grim (Tasmania, 40.7°S, 144.7°E), two samples from the Aurora Basin North firn air sampling site (Antarctica, 71.2°S, 111.4°E), and in situ monthly means since February 2017 from Aspendale, Victoria, Australia (38.0°S, 145.1°E), with uncertainty bars for Aspendale omitted to aid visual clarity. Growth rates and emissions are globally averaged. Emissions (c) are shown with units on the left y-axis and as $CO_2$-equivalent emissions based on a global warming potential of 12 000 (100-yr time frame), with units on the right y-axis. USA factory-level emissions are from URL: https://www.epa.gov/ghgreporting, accessed January 2019. The early history is shown as dotted lines to emphasize the greater uncertainties before 1978.

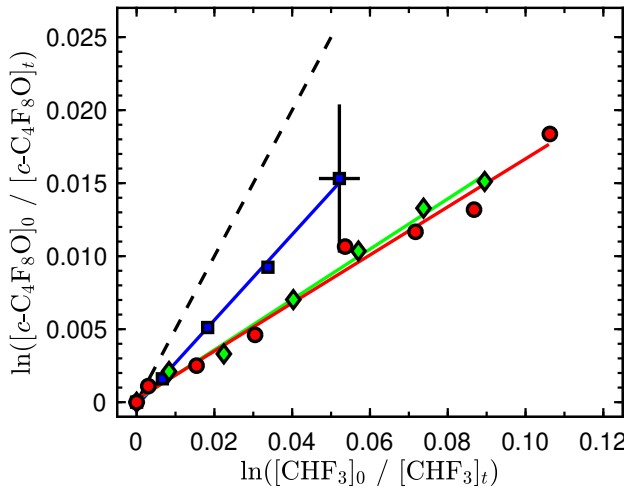

**Figure 3.** Relative rate data obtained in this work for the O($^1$D) + $c$-C$_4$F$_8$O reaction at 296 K. The different symbols are results from independent experiments and the solid lines are linear least-square fits to the data of the individual experiments. Representative estimated error bars from the infrared spectral analysis are included only on a single data point for improved clarity of the graph. The dashed line represents the upper-limit rate coefficient ratio recommended in this work.

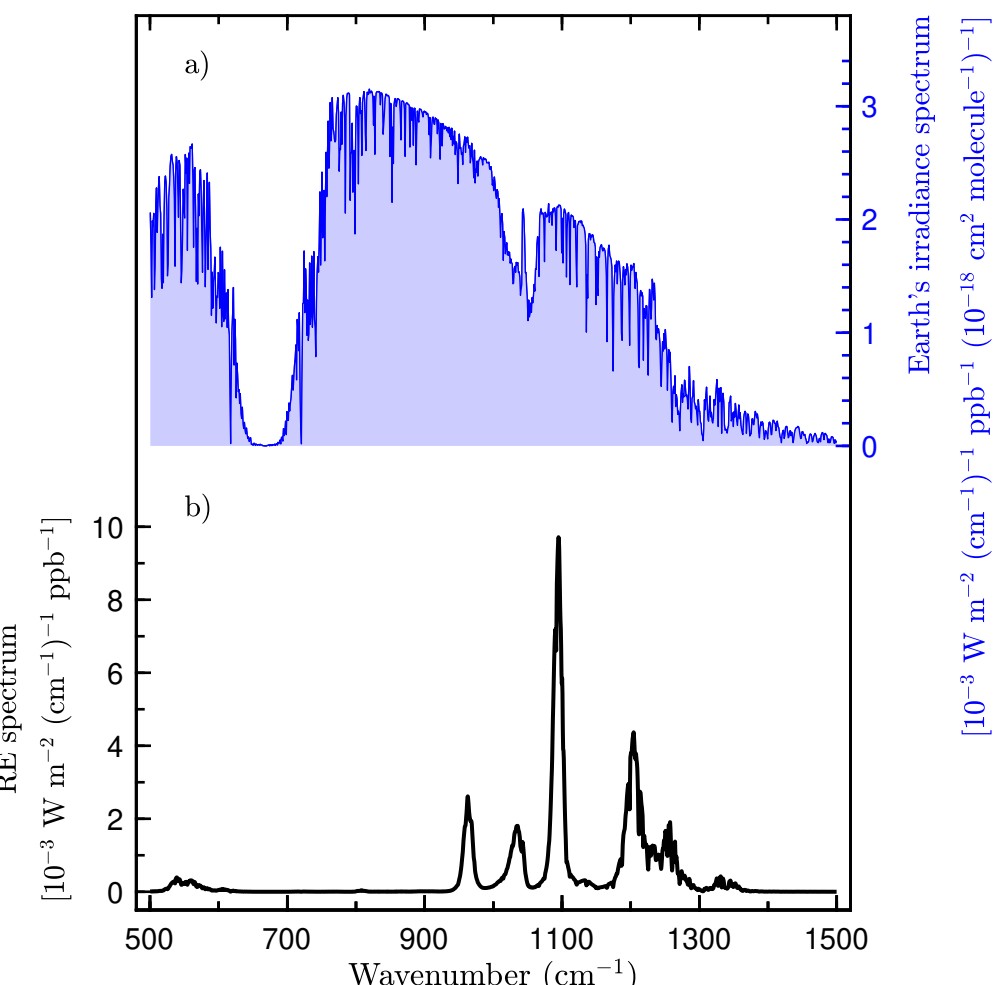

**Figure 4.** a) Earth's irradiance spectrum as reported by Hodnebrog et al. (2013) and b) radiative efficiency (RE) spectrum of octafluoroox-olane ($c$-$C_4F_8O$).