# Peer review of "Abundances, emissions, and loss processes of the long-lived and potent greenhouse gas octafluorooxolane (octafluorotetrahydrofuran, c-C4F8O) in the atmosphere"

_Atmospheric Chemistry and Physics, 2018_

## Referee Comment (RC1) · Anonymous Referee #1 · 16 Oct 2018

This paper presents atmospheric measurements of C4F8O along with laboratory studies of its IR spectrum and loss processes via photolysis and reaction with O(1D). The conclusion is that C4F8O is very long lived (> 3000 y) and has a GWP100 of the order of 12000.

Whereas the laboratory studies of the IR spectrum appear to be of high quality, the investigation of the UV spectrum and the reaction with O(1D) are rather disappointing. It is not clear if C4F8O absorbs in the 200-225 nm investigated (the authors do not present a Figure) and the O(1D) reaction appears to have been carried out with

a reference reactant that reacts too rapidly with O(1D) to get a good number for the rate coefficient. The most disturbing aspect is that the authors calculate that Lyman-alpha absorption may actually dominate the atmospheric loss of C4F8O, but only an unfounded guesstimate of the cross-section is made. A measurement of the VUV spectrum of C4F8O would significantly improve the laboratory studies and the manuscript as a whole. In addition, no mention is made of loss via hydrolysis (ocean) or reaction on particles. If such loss processes can indeed be neglected, the authors should provide physical constants (e.g. solubility / hydrolysis rate constants from the literature) that support this. These issues should be addressed in a revised version, as should the comments below:

Acronyms: use of acronyms is fine if it improves readability. The term ABN appears only three times in the manuscript (the first time being its definition). I would encourage use of the full name Aurora Basin North (as used in Figure 1).

P6L29 Some of the terms (A, I, L) need to be italicised as in the equation they refer to.

P6P6 Estimated accuracy of 1 %. Please state how this value is calculated.

P7L15 What was the effective spectral resolution of the MC/CCD ?

P7L24 The chemical element O should not by italicised

P8L4 Why not use the terms k1a and k2a to define the rate coefficients for reactions 1a and 2a. Also in equation (II) ?

P8L3 replace "at times t0 and t" with "at times zero and t".

P8L11 The chemicals element (O) should not be italicised

P8L19 is the "dark" loss of C4F8O measured in the presence of O3 ?

P8L23 Figure 2 is mentioned before Figure 1. The numbering of the Figures needs adjusting.

[Figure]

P8L28 Equation (I) describes absorption at any discrete wavelength, ïĄň and not over a range of wavelengths as the integrated band strength is calculated.

P8L28 The uncertainty estimates need attention. How do the individual uncertainties of combine to result in 3% ? How about the 1 % uncertainty quoted previously for the dilution in He ?

P9L3 cross sectionS decreasing monotonically. . ..

P9L7 An upper limit for the absorption cross section is presented. Please add a Figure (Supp. Inf) to clarify how this value was obtained.

P9L16 The problem with this measurements of the relative rate coefficient is that while measureable amounts (up to 10 %) of $CHF_3$ are removed, the changes in the $C_4F_8O$ concentration are too low to measure reliably. Could this have been improved by a better choice of reference compound (i.e. one that reacts more slowly with O(1D) ? Why was the experiment stopped after only 10% of $CHF_3$ was depleted ? Which absorption bands of $C_4F_8O$ and $CHF_3$ were used to derive the fractional losses ? Unless there are good arguments against, the authors should consider doing further experiments to nail down this number. Alternatively, they might consider using the correlation between ionisation potential and O(1D) rate coefficient that is frequently used to estimate the latter.

P9L24 The authors state that reaction with OH will not represent a loss of $C_4F_8O$ in the atmosphere. I agree, but the authors should state why this is most likely to be the case. Will a perfluorinated furan react with OH like other fully fluorinated organics ? What upper limit to the OH-rate coefficient would be necessary for OH reaction to compete with O(1D) induced losses ?

P10L3 The authors state that is reasonable to assume a Lyman-alpha cross-section of about $1x 10^{-17}$ as it is "roughly consistent" with highly fluorinated compounds. As the authors go on to conclude that this process has the shortest associated lifetime,

I find this unacceptably vague. What is the physical basis for assuming that a fluorinated furan will absorb at 121.6 nm with the same cross section as non-heterocyclic, perfluorinated gases ? Measurement of the cross section at this wavelength is not impossible. The NOAA lab certainly has VUV capability (e.g. for 185 nm measurements using Hg-lines) which could be extended to 121.6 nm. Surely even a rough experiment is better than a precarious assumption.

———————————————————

---

## Referee Comment (RC2) · Anonymous Referee #2 · 25 Oct 2018

General comments:

The article presents a large dataset and budget estimate for a newly detected compound in the atmosphere: c-C$_4$F$_8$O. Although its abundance is small (less than 0.1ppt) its radiative efficiency is strong and lifetime likely very high. It is still unregulated and sometimes viewed as a promising compound in terms of industrial applications (see for example Kočišek et al., 2018). I think that its scope and novelty make it adequate for a publication in ACP. I have some comments on the methodology and presentation.

[Figure]

The Northern Hemisphere (NH) measurements are little described and commented (p 1 l5-6, p3 l14-15). The article should explain how the North Hemisphere trend (dashed line on Figure 1) was constrained and evaluate the uncertainty on emissions resulting from the lack of NH constraints.

Similarly, the mixing ratio and emission trends between 1950 and 1978 are mostly constrained by a single firn air data point undergoing a large age distribution, and having a mixing ratio (6 ppq, Table S4) very close to the detection limit (5 ppq, p4 l13). The article should explicitly discuss the constraints on the anthropogenic versus natural sources of c-$C_4F_8O$, as well as the little constrained early emissions.

However, for a well-mixed very long lived species, a reasonable estimate of global emissions can be obtained from a simple one box model calculation. Presenting this simple calculation and comparing it to the elaborate approach used would improve the description of the main uncertainties and be helpful to non-specialist readers.

A first estimate of the lifetime of c-$C_4F_8O$ is provided but some important assumptions should be better described: the basis for the estimated Lyman-$\alpha$ lifetime and OH reactivity (comparison with species having similar bonding structures?), the possible role of other unexplored sinks such as surface loss (to ocean and land) and heterogeneous processes should be discussed at least in terms of perspectives.

Specific comments:

p2 l10-11: The Californian regulation could be mentioned (https://ww2.arb.ca.gov/resources/documents/semiconductor-regulation)

p5 l11-14: As pumping out the interstitial air from deep firn can be difficult and induce contamination, more indications should be provided about the multi-species consistency of model results for the deep firn air sample used and the overall firn. For example, the RMSD/$\sigma$ indicator used in Buizert et al. (2012) could be provided. The reason why so few depth levels were analyzed for c-$C_4F_8O$ should be given, sample

size issue?

p5 l14-18: The Trudinger et al. (2013) model uses both molecular and eddy diffusivity terms. As this has the same effect as modifying the diffusion coefficient, the relative roles of molecular and eddy diffusivity terms for the ABN firn should be commented.

p5 l19-26: how were the North Hemisphere concentrations evaluated?

p5 l29: Vollmer et al. (2016, 2018) used multi-depths firn air constraints from both hemispheres. The methodological adaptations to the lack of NH constraints should be described.

p6 l1: I do not understand what the Green's functions from the 12-box model are and did not see an explanation in Vollmer et al. (2016, 2018)

p6 l11-13: The emission values in Ivy et al. (2012) start in 1980 (Table 3), how was the prior estimate designed for the 1950-1980 period and what impact does it have on the final solution for this weakly constrained period?

p7 l23 and after, including section 2.2 of the Supplement: a single notation should be adopted to name reaction rates, avoid using $k_R$, then $k_1$ (implicit) and $k_2$, then $k_{c-C_4F_8O}$.

p10 l9: Figure 4 is little commented, it could be shifted to the Supplement or combined with Fig. 2

p10 l26-27: circular argument, the calculated growth rate is small because the measured concentration trend is weak (in recent years), not the contrary.

p11 l6-9: the important Aspendale dataset (thousands of measurements) is briefly summarized in Table S3 and very briefly commented. A more in-depth discussion of c-$C_4F_8O$ variability at various sub-annual time scales and recent trend, as well as a plot (at least in the Supplement) would be useful.

p11 l26: the wording "a few other synthetic greenhouse gases" implicitly assumes that c-$C_4F_8O$ is purely anthropogenic but this is not discussed in the article

p12 l6: due to the high cost of Antarctic field operations, research programs and logistic institutions financing them are usually explicitly named.

pS7 l16 of the Supplement: the chosen 1 ppq uncertainty seems small compared to the stdv values in Table S3, this choice should be further commented.

pS10 l6-10 of the Supplement: this discussion of background / non anthropogenic level of c-$C_4F_8$O should be in the main article

Technical corrections:

p 5 l7 use indices in c-C4F8O

p 5 l11-14 repeats l4-8

p 5 l16 and 18 suppress ))

p 6 l12 from observations of

New references:

Kočišek, J, Janečková, R. and Fedor, J., Long-lived transient anion of c-$C_4F_8$O, Journal of Chemical Physics, 148(7), 074303, 2018.

Buizert et al., Gas transport in firn: multiple-tracer characterisation and model inter-comparison for NEEM, Northern Greenland, Atmos. Chem. Phys., 12, 4259-4277, https://doi.org/10.5194/acp-12-4259-2012, 2012.
* * *

---

## Author Comment (AC1) · 10 Jan 2019

By Anonymous Referee #1 ()

Replies to the reviewer comments are added in blue color following each comment, and the revised text is in green.
We thank the reviewer for his/her input. We believe that with the suggested changes to these valuable comments, the manuscript will improve.

Reviewer Comment: This paper presents atmospheric measurements of C4F8O along with laboratory studies of its IR spectrum and loss processes via photolysis and reaction with O(1D). The conclusion is that C4F8O is very long lived (> 3000 y) and has a GWP100 of the order of 12000.

Whereas the laboratory studies of the IR spectrum appear to be of high quality, the investigation of the UV spectrum and the reaction with O(1D) are rather disappointing. It is not clear if C4F8O absorbs in the 200-225 nm investigated (the authors do not present a Figure) and the O(1D) reaction appears to have been carried out with a reference reactant that reacts too rapidly with O(1D) to get a good number for the rate coefficient. The most disturbing aspect is that the authors calculate that Lyman-alpha absorption may actually dominate the atmospheric loss of C4F8O, but only an unfounded guesstimate of the cross-section is made. A measurement of the VUV spectrum of C4F8O would significantly improve the laboratory studies and the manuscript as a whole. In addition, no mention is made of loss via hydrolysis (ocean) or reaction on particles. If such loss processes can indeed be neglected, the authors should provide physical constants (e.g. solubility / hydrolysis rate constants from the literature) that support this. These issues should be addressed in a revised version, as should the comments below:

Reply: We address the individual concerns below where most of them are mentioned in a more specific form. With regard to losses other than atmospheric, we refer to the replies to reviewer 2, who has commented on similar issues. Note that research on this compound is extremely sparse, e.g. we have not found reasonable peer-reviewed solubility data, to further investigate other potential sink. This is also clearly out of the scope of this study.

Reviewer Comment: Acronyms: use of acronyms is fine if it improves readability. The term ABN appears only three times in the manuscript (the first time being its definition). I would encourage use of the full name Aurora Basin North (as used in Figure 1).

Reply: We agree with this suggestion and suggest to use the full name for Aurora Basin North in the main text. However, in the supplement, the term 'Aurora Basin North' appears many times, consequently we suggest to use the abbreviated form (ABN) there.

Reviewer Comment: P6L29 Some of the terms (A, I, L) need to be italicised as in the equation they refer to.

Reply: This is now fixed, also with regard to equation II.

Reviewer Comment: P6P6 Estimated accuracy of 1 %. Please state how this value is calculated.

Reply: This probably refers to P7L6, not P6L6. The estimated accuracy is based on the estimated uncertainty of the pressure measurements used in the preparation of the standard gas mixtures. These include the uncertainty in the sample pressure, the total mixture pressure, and the linearity of the pressure gauge (estimated to be 0.2%). On the basis of these estimates, we report an estimated accuracy of around 1%.

We suggest to revise the sentence on P7L5 "The dilute mixtures of $c$-$C_4F_8O$ in a He (UHP, 99.999 %) bath gas was prepared manometrically in a 12 L Pyrex bulb with an estimated accuracy of 1 %." by "The dilute mixtures of $c$-$C_4F_8O$ in a He (UHP, 99.999 %) bath gas were prepared manometrically in a 12 L Pyrex bulb with an estimated accuracy of 1%, as derived from the accuracy of the pressure measurements. The pressure measurement uncertainty includes the uncertainty in the sample pressure, the total mixture pressure, and the linearity of the pressure gauge (estimated to be 0.2%)."

Reviewer Comment: P7L15 What was the effective spectral resolution of the MC/CCD?

Reply: The resolution of the UV spectrometer was ~1 nm.
Actions Taken: P7L16, the sentence "…using a 0.5 m spectrometer equipped with a charge-coupled device (CCD) detector." will be replaced by "… using a 0.5 m spectrometer (1 nm resolution) equipped with a charge-coupled device (CCD) detector."

Reviewer Comment: P7L24 The chemical element O should not by italicised.

Reply: This is now fixed.

Reviewer Comment: P8L4 Why not use the terms k1a and k2a to define the rate coefficients for reactions 1a and 2a. Also in equation (II) ?

Reply: Yes, $k_{1a}$ and $k_{2a}$ are the reactive rate coefficients for the reaction of $O(^1D)$ with $c$-$C_4F_8O$ and $CHF_3$, respectively. In the Equation (II), $k_{c\text{-}C4F8O}$ and $k_{CHF3}$ are suggested as replacements for $k_{1a}$ and $k_{2a}$, respectively.

Reviewer Comment: P8L3 replace "at times t0 and t" with "at times zero and t".

Reply: Done, it now reads 'at times zero ($t_0$) and $t$.

Reviewer Comment: P8L11 The chemicals element (O) should not be italicised

Reply: This is now fixed.

Reviewer Comment: P8L19 is the "dark" loss of C4F8O measured in the presence of O3 ?

Reply: Yes, under these experimental conditions, the loss of $c$-$C_4F_8O$ in the absence of $O(^1D)$ production implies the dark loss of $c$-$C_4F_8O$ under identical experimental conditions.
Actions Taken: The sentence "Experiments performed separately demonstrated that there was no significant loss of $c$-$C_4F_8O$ or $CHF_3$ in the absence of $O(^1D)$ production." will be replaced by "Experiments performed separately demonstrated that there was no significant loss of $c$-$C_4F_8O$ or $CHF_3$ under identical conditions in the absence of photolysis."

Reviewer Comment: P8L23 Figure 2 is mentioned before Figure 1. The numbering of the Figures needs adjusting.
Reply: This is now fixed, first-mentioned figure (IR spectrum) is now Figure 1.

Reviewer Comment: P8L28 Equation (I) describes absorption at any discrete wavelength, ï A̮n and not over a range of wavelengths as the integrated band strength is calculated.

Reply: The reviewer is correct, and the revision is suggested for P8L27 by replacing the sentence "Absorption spectra were quantified using Beer's law" by "Absorption spectra, A(v) (base e), (or integrated band strengths) were quantified using Beer's law"

Reviewer Comment: P8L28 The uncertainty estimates need attention. How do the individual uncertainties of combine to result in 3% ? How about the 1 % uncertainty quoted previously for the dilution in He ?

Reply: The absolute uncertainty of the total integrated band strength was based on the root mean square analysis including the estimated uncertainties of the measured absorbance (±0.5%), the optical pathlength (±0.5%) and the concentration (±1%). The uncertainty of the concentration is directly related to the pressure uncertainty where a value of ±1% has been used for the calculation. This resulted to a total uncertainty of 2.1%. We assigned a conservative uncertainty of 3%. We suggest to leave the text unaltered and not include such detail of how the estimate was derived.

Reviewer Comment: P9L3 cross sectionS decreasing monotonically

Reply: This is now fixed, the word is pluralized.

Reviewer Comment: P9L7 An upper limit for the absorption cross section is presented. Please add a Figure (Supp. Inf) to clarify how this value was obtained.

Reply: The following figure will be added in the Supplement. At the end of P9L4, the sentence "(An example of the measured UV spectrum of the $c$-$C_4F_8O$ sample is shown in the Supplement)" will be added.

[Figure]

Figure S2. Measured 296 K gas-phase UV spectrum of the octafluorotetrahydrofuran ($c$-$C_4F_8O$) sample used in this study. The $c$-$C_4F_8O$ concentration was 2.16□ $\times 10^{19}$ molecule $cm^{-3}$. Assuming no absorption from possible sample impurities, the $c$-$C_4F_8O$ absorption cross section at 210 nm was found to be 6.25 x $10^{-24}$ $cm^2$ molecule$^{-1}$. As discussed in Section 3.2, such a low cross section is susceptible to overestimation due to the presence of impurities in the $c$-$C_4F_8O$ sample. Therefore, this spectrum was considered as an upper-limit in our lifetime analysis.

Reviewer Comment: P9L16 The problem with this measurements of the relative rate coefficient is that while measureable amounts (up to 10 %) of CHF3 are removed, the changes in the C4F8O concentration are too low to measure reliably. Could this have been improved by a better choice of reference compound (i.e. one that reacts more slowly with O(1D))? Why was the experiment stopped after only 10% of CHF3 was depleted ? Which absorption bands of C4F8O and CHF3 were used to derive the fractional losses? Unless there are good arguments against, the authors should consider doing further experiments to nail down this number. Alternatively, they might consider using the correlation between ionisation potential and O(1D) rate coefficient that is frequently used to estimate the latter.

Reply: Regarding the $O(^1D)$ reaction, presently there is not a more reliable reference compound to be used. Note that it is extremely difficult to accurately measure small $O(^1D)$ rate coefficients. It is also worth noting that an improved measure of the $O(^1D)$ reactive rate coefficient for the furan reaction would not significantly alter our conclusions regarding the atmospheric loss of the furan. In this work, a conservative upper-limit rate coefficient was reported that can be refined in the future, if warranted.
**Regarding the extent of reaction:** The 248 nm photolysis of ozone was used as a source of $O(^1D)$ in the photochemical reactor. The introduction of ozone to the reactor was accompanied by an addition of molecular oxygen. After multiple ozone additions, the reaction of $O(^1D)$ by $O_2$ can dominate the $O(^1D)$ loss, making the $O(^1D)$ + furan and $O(^1D)$ + $CHF_3$ reactions less significant.
**Regarding the absorption bands:** The absorption bands used in this study were 1120-1000 and 1180-1120 $cm^{-1}$ for $c$-$C_4F_8O$ and $CHF_3$, respectively.
**Regarding further laboratory studies of the $O(^1D)$ reaction:** Based on our experimental data and analysis, the $O(^1D)$ reaction will have a minor impact on the furan global atmospheric lifetime. Therefore, at our current level of understanding of the atmospheric chemistry and modeling (i.e., lifetime determination for persistent compounds) further refinement of the $O(^1D)$ rate coefficient is not warranted at this time.

The sentence "the precision of the infrared spectral subtractions are the primary sources of uncertainty in the measurements." will be replaced by "the precision of the infrared spectral subtractions are the primary sources of uncertainty in the measurements. The low conversion of the $c$-$C_4F_8O$ and $CHF_3$ achieved in these experiments was primarily due to the build up of $O_2$ associated with the addition of ozone to the reactor, making the loss of $O(^1D)$ by reaction with $O_2$ more significant than its reaction with $c$-$C_4F_8O$ and $CHF_3$. The absorption bands used in this study were 1120-1000 and 1180-1120 $cm^{-1}$ for $c$-$C_4F_8O$ and $CHF_3$, respectively."

P9L16, at the end of the paragraph of the Section 3.3, the sentence "Given the small $O(^1D)$ rate coefficient for the reaction between $O(^1D)$ and $c$-$C_4F_8O$, a refinement of the measured $O(^1D)$ reactive rate coefficient will have a negligible impact to the total atmospheric loss of $c$-$C_4F_8O$." is suggested to be added.

Reviewer Comment: P9L24 The authors state that reaction with OH will not represent a loss of C4F8O in the atmosphere. I agree, but the authors should state why this is most likely to be the case. Will a perfluorinated furan react with OH like other fully fluorinated organics? What upper limit to the OH-rate coefficient would be necessary for OH reaction to compete with O(1D) induced losses?

Reply: As stated, the loss of $c$-$C_4F_8O$ by reaction with OH radicals is most likely a negligible process. Due to laboratory challenges, it is not possible to measure values approaching $1 \times 10^{-17}$ $cm^3$ molecule$^{-1}$ s$^{-1}$ (equivalent to 3000 years lifetime). We estimate that our laboratory measurements would, at best, yield a rate constant upper-limit of ~$1 \times 10^{-14}$ $cm^3$ molecule$^{-1}$ s$^{-1}$, which would be a gross overestimate of the true rate constant. Reporting such a rate constant would be very misleading and possibly misinterpreted by the community. We did not attempt to conduct OH reactivity measurements of this compound for these reasons. The expected lowreactivity of the furan is supported by its low O(1D) reactivity. To compete with O(1D) induced losses, the OH rate coefficient would be equivalent to $>1 \times 10^{-18}$ $cm^3$ $molecule^{-1}$ $s^{-1}$.

Actions Taken:  P9L22, the sentence "The loss of $c$-$C_4F_8O$ via reaction with the OH radical is assumed to make a negligible contribution to the global lifetime and consequently we ignore the last term in Equation 4." is planned to be replaced by "The loss of c-$C_4F_8O$ via the reaction with the OH radical is assumed to make a negligible contribution to the global lifetime in our analysis. the OH rate coefficient would need to be ~$1 \times 10^{-17}$ $cm^3$ $molecule^{-1}$ $s^{-1}$ (equivalent to a 3000 year lifetime) to make a significant global lifetime contribution.  Such a low rate coefficient represents a significant challenge to current rate coefficient measurement methods.  Additionally, an expected OH low-reactivity of the furan is supported by its low O($^1$D) reactivity measured in this work. Additional laboratory studies that are beyond the scope of the present work would be needed to quantify the OH reaction."

Reviewer Comment: P10L3 The authors state that is reasonable to assume a Lyman-alpha cross-section of about 1x 10-17 as it is "roughly consistent" with highly fluorinated compounds. As the authors go on to conclude that this process has the shortest associated lifetime, I find this unacceptably vague. What is the physical basis for assuming that a fluorinated furan will absorb at 121.6 nm with the same cross section as non-heterocyclic, perfluorinated gases? Measurement of the cross section at this wavelength is not impossible. The NOAA lab certainly has VUV capability (e.g. for 185 nm measurements using Hg-lines) which could be extended to 121.6 nm. Surely even a rough experiment is better than a precarious assumption.

Reply:  The Lyman-alpha absorption cross section was not measured as part of the present study. We are implying that it should be measured in future studies, which would refine our analysis.  In the absence of experimental data, we surveyed available cross section data for highly fluorinated compounds and found that a cross section of $1 \times 10^{-17}$ $cm^2$ $molecule^{-1}$ would represent a reasonable cross section upper-limit.  Using this cross section value in our lifetime analysis provides a Lyman-alpha photolysis lifetime of ~4500 years.  Assuming a smaller or greater cross section would yield slightly longer or shorter lifetimes, although the relationship is not linear due to the dependence on the transport time to the mesosphere.  Therefore, in the absence of experimental data, we believe our assumed cross section and lifetime is reasonable. Laboratory studies to measure VUV cross sections would help refine our analysis but not alter the primary conclusion from this work that the atmospheric lifetime of the furan is extremely long.  Note that measuring VUV spectra in the laboratory is significantly different than measuring 185 nm cross sections and requires specialized equipment.

Actions Taken: P10L2, the sentences "It is reasonable to assume a Lyman-α cross section of $10^{-17}$ $cm^2$ $molecule^{-1}$ for $c$-$C_4F_8O$ which would be roughly consistent with values for highly fluorinated compounds (SPARC, 2013).  The estimated lifetime due to Lyman-α photolysis, $\tau_{Lyman-\alpha}$, is then ~ 4500 years (a smaller Lyman-α cross section would lead to a longer lifetime)." will be replaced by "The scope of the present study did not include a measurement of the Lyman-α cross section.  It is however reasonable to assume a Lyman-α cross section of $10^{-17}$ $cm^2$ $molecule^{-1}$ for $c$-$C_4F_8O$ which is in the range of values for highly fluorinated compounds, $(0.035 - 10) \times 10^{-17}$ $cm^2$ $molecule^{-1}$ (SPARC, 2013).  Therefore, in the absence of experimental data, we consider the Lyman-α cross section, $1 \times 10^{-17}$ $cm^2$ $molecule^{-1}$, used in our lifetime analysis as a reasonable estimate.  Note that a smaller (larger) Lyman-α cross section would lead to a longer (shorter) photolysis lifetime, although the lifetime dependence on the cross section value is not linear due to the lifetime dependence on the transport time to the mesosphere."

---

## Author Comment (AC2) · 10 Jan 2019

By Anonymous Referee #2 ()

Replies to the reviewer comments are added in blue color following each comment, and the revised text is in green.
We thank the reviewer for his/her input. We believe that with the suggested changes to these valuable comments, the manuscript will improve.

Reviewer Comment: General comments:
The article presents a large dataset and budget estimate for a newly detected compound in the atmosphere: $c$-C$_4$F$_8$O. Although its abundance is small (less than 0.1ppt) its radiative efficiency is strong and lifetime likely very high. It is still unregulated and sometimes viewed as a promising compound in terms of industrial applications (see for example Kočišek et al., 2018). I think that its scope and novelty make it adequate for a publication in ACP. I have some comments on the methodology and presentation.

Reviewer Comment: The Northern Hemisphere (NH) measurements are little described and commented (p 1 l5-6, p3 l14-15). The article should explain how the North Hemisphere trend (dashed line on Figure 1) was constrained and evaluate the uncertainty on emissions resulting from the lack of NH constraints.

Reply: Northern Hemisphere samples are planned to be better described in the Supplement as follows:
"S-2.5 Northern Hemisphere (NH) samples
The Southern Hemisphere Cape Grim Air Archive (CGAA) samples were complemented with a few archived air samples from the Northern Hemisphere (see also Table S3). Some of these samples were collected as whole air ambient background samples for original calibration purposes: UAN920470 at Cape Meares, Oregon, most likely cryogenic techniques; T-EMPA-1 and J-187 at La Jolla, California using an oil-free diving compressor (Rix Industries); EG-003 at Jungfraujoch, Switzerland, using cryogenic techniques. H-160 at Mace Head, Ireland, using an oil-free diving compressor. These samples were all collected in internally electropolished stainless steel canisters (Essex Industries, USA). Two samples collected at Dubendorf (DUE161216-D2 and DUE161216) were collected into 6-L internally electropolished cylindrical custom-fabricated containers (LabCommerce, California) using a diaphragm pump (KNF-N-022-ANE, Neuberger), for the specific purpose of this project. These two samples, and EG-003 and H-160 were shipped from Empa to CSIRO for analysis along with the CGAA samples under same measurement conditions."

We have now also better described how the NH trend was constrained. The text added is:
"The $c$-C$_4$F$_8$O measurements in the Southern Hemisphere provide a strong constraint on the trend in both hemispheres due to the very long lifetime of $c$-C$_4$F$_8$O in the atmosphere, the relatively rapid mixing of the atmosphere, and the expectation that most $c$-C$_4$F$_8$O emissions are in the Northern Hemisphere (NH). Most anthropogenic gases are released predominantly in the NH, including gases released by the semiconductor industry. The assumption of mainly NH emissions for $c$-C$_4$F$_8$O leads to higher mole fraction values in the NH, and this is confirmed by comparison of the modelled NH history with the few NH measurements that we do have. In the Supplement, we test the sensitivity of inferred global emissions and mole fraction in both hemispheres to the assumed spatial distribution of emissions, and show the uncertainty in inferred mole fraction in both hemispheres."

Reviewer Comment: Similarly, the mixing ratio and emission trends between 1950 and 1978 are mostly constrained by a single firn air data point undergoing a large age distribution, and having a mixing ratio (6 ppq, Table S4) very close to the detection limit (5 ppq, p4 l13). The article should

explicitly discuss the constraints on the anthropogenic versus natural sources of c-$C_4F_8O$, as well as the little constrained early emissions.

Reply: We agree, and suggest to change the text in the results section to address these concerns: "This suggests that $c$-$C_4F_8O$ was below 10 ppq in the Southern Hemisphere before 1978. However it is impossible to further pin down the first appearance of this compound in the atmosphere and the exact course of the abundance until ~1980 because our knowledge of $c$-$C_4F_8O$ prior to the CGAA is based on only one firn sample measurement with air spanning several decades (see calculated Green's functions in the Supplement). Also, potential small contamination during firn air sampling by modern air or sampling devices cannot be fully excluded. Additionally, the measurement of the older firn air sample is close to the instrument's detection limit. Given these limitations, we are not able to draw any conclusions on any potential naturally-occurring $c$-$C_4F_8O$. Nevertheless, the two firn air sample measurements allow us to draw conclusions on storage stability of $c$-$C_4F_8O$ in canisters. Given that the storage time of the two firn air samples in the canisters is much shorter than those of the older CGAA samples, the good agreement of the firn air results with those of the CGAA is supportive of storage stability of $c$-$C_4F_8O$ in the CGAA tanks and confirms that the observed multidecadal record is not a simple artifact of degradation of $c$-$C_4F_8O$ in canisters over time."

In addition we suggest to show the early history with dotted lines rather than solid lines,to emphasize the greater uncertainty before 1978. See revised figure further below.

Reviewer Comment: However, for a well-mixed very long lived species, a reasonable estimate of global emissions can be obtained from a simple one box model calculation. Presenting this simple calculation and comparing it to the elaborate approach used would improve the description of the main uncertainties and be helpful to non-specialist readers.

Reply: We believe that adding another model calculation would be rather confusing than helping to understand the main uncertainties. The 12-box model has been used and validated many times in the past in numerous publications and doesn't need to be re-assessed here. Nevertheless, we have made a quick intercomparison based on an 1-box model using the following assumption: 1.8E20 mol of air in the total atmosphere; well mixed (no delay in stratosphere); no sinks; using the fitted observations from the 12-box model. Admittedly, the latter assumption creates some degree of dependency to the 12-box model, but an independent fit through the observations would not significantly alter the results. The result is shown here in the subfigure c) as orange line. Some of the discrepancies to the emissions from the 12-box model is likely caused by the above assumption, in particular that of a uniform vertical atmosphere. In addition, we have calculated the cumulative emission in a 1-box model approach using the end-of record mole fractions of about 74.5 ppq. Again using 1.8 E20 mol of air in the total atmosphere and no sinks for $c$-$C_4F_8O$, we calculate 2.89 kt, which compares well with the 2.85 kt from the 12-box model. We propose to not mention these 1-box model results in the revised text for the reasons mentioned above.

Reviewer Comment: A first estimate of the lifetime of c-$C_4F_8O$ is provided but some important assumptions should be better described: the basis for the estimated Lyman-α lifetime and OH reactivity (comparison with species having similar bonding structures?), the possible role of other unexplored sinks such as surface loss (to ocean and land) and heterogeneous processes should be discussed at least in terms of perspectives.

Reply: Thank you for addressing these points. We handle the comments regarding Lyman-alpha and OH reactivity as part of the replies to reviewer 1, and would like to refer to that reply.
With regard to other possible sinks, we suggest to add the following sentence to the Introduction (following the discussion on the lack of existing information on atmospheric loss of the compound): "Information is also lacking on other potential loss processes for $c$-$C_4F_8O$, such as uptake by oceans and land".

We also suggest to add the following sentence to the description of the 12-box model: "Loss processes other than those in the atmosphere, such as uptake by land and oceans, and potential natural sources, are not included in the model."

Further, we suggest to add the following sentences to the discussion of the 'Atmospheric Lifetime (3.4)':
"This study has focused primarily on the atmospheric loss processes of $c$-$C_4F_8O$, i.e., potential deposition or heterogeneous loss processes of $c$-$C_4F_8O$ were beyond the scope of this study. Deposition or heterogeneous loss processes, if significant, would lead to a shorter global lifetime for $c$-$C_4F_8O$."

Ultimately we modified the last sentence in the conclusions to: "However, even if emissions were completely halted, it will, under the assumption of insignificant non-atmospheric sinks, take thousands of years for $c$-$C_4F_8O$ to be removed from the atmosphere."

Reviewer Comment: Specific comments:

p2 l10-11: The Californian regulation could be mentioned
(https://ww2.arb.ca.gov/resources/documents/semiconductor-regulation)
Reply: We have explored this a bit more. Rather than mentioning the Californian regulations, we have mentioned the US EPA regulations and IPCC. The revised text is suggested as

"The compound is listed in the Intergovernmental Panel on Climate Change (IPCC) 2006 guidelines in support of UNFCCC (IPCC, 2006, Volume 1, Chapter 8) as a compound, for which countries are encouraged to provide emissions estimates (on a mass unit until a published greenhouse warming potential (GWP) will become available). In the 2013 Revisions of the UNFCCC reporting guidelines (UNFCCC 2013), $c$-$C_4F_8O$ is absent from the list of compounds with mandatory reporting. Additional reporting regulations exist on country or state levels. In the USA, large suppliers and emitters of $c$-$C_4F_8O$ are required to report the amounts they supply or emit under the Greenhouse Gas Reporting Program (GHGRP, URL: https://www.epa.gov/ghgreporting, accessed January 2019). When CO2-equivalent emissions are required for these submissions, a default GWP for fully fluorinated GHGs of 10,000 is used due to the lack of a peer-reviewed GWP. Emissions have mainly been reported under the ``Fluorinated Gas Production'' subpart for 2011--2017 with a maximum of 40 t in 2013 and a subsequent decline to 4.5 t by 2017"

Reviewer Comment: p5 l11-14: As pumping out the interstitial air from deep firn can be difficult and induce contamination, more indications should be provided about the multi-species consistency of model results for the deep firn air sample used and the overall firn. For example, the RMSD/σ indicator used in Buizert et al. (2012) could be provided. The reason why so few depth levels were analyzed for c-$C_4F_8O$ should be given, sample size issue?

Reply: The two ABN firn measurements play only a small role in this study. It is discussed above that the early history is not well constrained by the single deep firn measurement due to its age spread, and that contamination cannot be ruled out. Another publication is underway that will describe the ABN measurements and modelling in much greater detail, including showing how well the firn model fits all measurements used for calibration. The level of detail suggested by the reviewer is not seen as necessary for this study given the small role of the firn measurements, so no further change has been made.

We suggest to add the following sentence to explain why there were only 2 samples available for this project (in Methods).
"Only two samples were available for the present study as other samples from this site were used for a different halocarbon study."

Reviewer Comment p5 l14-18: The Trudinger et al. (2013) model uses both molecular and eddy diffusivity terms. As this has the same effect as modifying the diffusion coefficient, the relative roles of molecular and eddy diffusivity terms for the ABN firn should be commented.

Reply: We plan to comment on this by adding the following: "Only molecular diffusion was used for the ABN firn model calculations; eddy diffusivity is sometimes used in the deep firn but was not used here as the parameters were not well constrained by the available measurements".

Reviewer Comment: p5 l19-26: how were the North Hemisphere concentrations evaluated?
Reply: We are not clear about this comment. We assume that it is related to the first comment about the NH concentrations, and believe to have addressed this comment sufficiently there.

Reviewer Comment: p5 l29: Vollmer et al. (2016, 2018) used multi-depths firn air constraints from both hemispheres. The methodological adaptations to the lack of NH constraints should be described.

Reply: The method was not adapted from Vollmer et al (2016, 2018) due to the use of only SH constraints. As described above, the NH trends are well constrained by SH measurements due to the long lifetime and predominantly NH emissions.

Reviewer Comment: p6 l1: I do not understand what the Green's functions from the 12-box model are and did not see an explanation in Vollmer et al. (2016, 2018)

Reply: To clarify this, we suggest to add the following text "Green's functions derived from the 12-box atmospheric model relate atmospheric mole fraction in the high-latitude Northern and Southern Hemispheres to annual global emissions in preceding years, and are used in the inversion (Trudinger et al;, 2016)."

Reviewer Comment: p6 l11-13: The emission values in Ivy et al. (2012) start in 1980 (Table 3), how was the prior estimate designed for the 1950-1980 period and what impact does it have on the final solution for this weakly constrained period?

Reply: We suggest to add some text to correct for this shortfall. Although of importance, we consider it a detail that better fits into the supplement, in particular in relation to (original) Fig. S4, where we graphically show what we did. The suggested text is:
"We construct a $c$-$C_4F_8O$ prior history from emissions of perfluorooctane because this compound has similarly low abundances and a long lifetime as $c$-$C_4F_8O$. For our standard case, we use perfluorooctane emissions published by Ivy et al. (2012) for the 1980 – 2010 period with the perfluorooctane 2010 value as a constant value for 2010 – 2017 and the 1980 value for perfluorooctane for the 1950 – 1980 period. We also test the sensitivity of our results to a number of other prior histories. a) the standard case doubled, b) the standard case halved, c) the standard case with emissions before 1980 extrapolated back to zero in 1950 and d) a small linearly increasing function (all shown in Fig. S4). Our analysis shows that the emissions derived for $c$-$C_4F_8O$ are rather insensitive to the choice of the prior, because the prior is used as a starting point for the inversion only, and not as a constraint."

Reviewer Comment: p7 l23 and after, including section 2.2 of the Supplement: a single notation should be adopted to name reaction rates, avoid using kR, then k1 (implicit) and k2, then kc-$C_4F_8O$.

Reply: We agree and suggest the following revision: "kR" will be replaced by "$k_1$", and "$k_{c\text{-}C4F8O}$" and "$k_{CHF3}$" will be replaced by $k_{1a}$ and $k_{2a}$, respectively in the Equation II. In the supplement, "$k/k_{CHF3}$" in Table S5 will be replaced by "$k_{1a}/k_{2a}$". In the footnote of Table S5, "$k_{CHF3}$ ($O(^1D) + CHF_3$) = $2.4 \times 10^{-12}$ cm$^3$ molecule$^{-1}$ s$^{-1}$" has been replaced by "$k_{2a} = 2.4 \times 10^{-12}$ cm$^3$ molecule$^{-1}$ s$^{-1}$".

Reviewer Comment: p10 l9: Figure 4 is little commented, it could be shifted to the Supplement or combined with Fig. 2

Reply: We prefer the Figure 4 to remain in the manuscript. It shows how IR absorption bands fall within the atmospheric window qualitatively and quantitatively. It is also used as a basis for Radiative Efficiency calculations and the discussions on GWP (p. 10 L. 9–13).

Reviewer Comment: p10 l26-27: circular argument, the calculated growth rate is small because the measured concentration trend is weak (in recent years), not the contrary.

Reply: Thank you for spotting this. We suggest to change the sentence(s) to: "The growth rate was at a maximum of 4.3 ppq/yr in 2004 and declined from that to <0.15 ppq/yr in 2017 as a consequence of the relatively constant abundances in the last few years."

Reviewer Comment: p11 l6-9: the important Aspendale dataset (thousands of measurements) is briefly summarized in Table S3 and very briefly commented. A more in-depth discussion of $c$-$C_4F_8O$ variability at various sub-annual time scales and recent trend, as well as a plot (at least in the Supplement) would be useful.

Reply: We plan to address this comment by providing an additional section in the supplement including a figure showing the high-resolution data set from in-situ measurements at Aspendale. There is no in-depth discussion on sub-annual time scales and trends, as there is no variability for this record, which we have already stated in the main text.
The revised text and figure is suggested as follows:

"S-2.6 In-situ measurements of $c$-$C_4F_8O$ at Aspendale
Regular measurements of $c$-$C_4F_8O$ in ambient air at Aspendale were started in February 2017. These were conducted on a 2-hourly basis whereas each air measurement is bracketed by standard measurements. Results are shown in Fig. S4. A few ambient air measurements were also made in late 2016 during the CGAA measurement phase. These were made from 3 L samples (vs the regular 2 L samples) and show improved precisions compared to the remaining record. The 2-year record shows constant $c$-$C_4F_8O$ mole fractions within the precisions of the measurements. There is no sign of any pollution events in this record suggesting that there are no significant sources of $c$-$C_4F_8O$ within the footprint of the site. Furthermore and given the long atmospheric lifetime of the compound, the absence of a significant trend is suggestive of the absence of major global emissions in the last years."

Figure S4. Ambient air measurements of c-C4F8O at Aspendale (Victoria, Australia, 38.0 °S, 145.1 °E). The measurements are expressed as dry air mole fraction in parts-per-trillion on the Empa-2013 calibration scale. Results show constant $c$-$C_4F_8O$ within the precision of the measurement.

Reviewer Comment: p11 l26: the wording "a few other synthetic greenhouse gases" implicitly assumes that $c$-$C_4F_8O$ is purely anthropogenic but this is not discussed in the article

Reply: The reviewer is correct. We suggest to remove the word 'synthetic'.

Reviewer Comment: p12 l6: due to the high cost of Antarctic field operations, research programs and logistic institutions financing them are usually explicitly named.

Reply: While we agree with the reviewer on the large-scale operations of the Antarctic field programs, this has to be also viewed in ratio to other contributions and be somewhat balanced. For example, compared to the two firn air samples, the input from the general AGAGE operation is large, AGAGE is a very large and costly long-term network and yet we cannot acknowledge all of its sub-contributions and funding agencies.

Reviewer Comment: pS7 l16 of the Supplement: the chosen 1 ppq uncertainty seems small compared to the stdv values in Table S3, this choice should be further commented.

Reply: The 1 ppq uncertainty is the magnitude of the uncertainty in the smoothed annual values, that come from the smoothing spline fit to the measurements. The uncertainties are much less than the stdv values of individual measurements, but relate to the uncertainty from the smoothing spline.

Reviewer Comment: pS10 l6-10 of the Supplement: this discussion of background / non anthropogenic level of $c$-$C_4F_8O$ should be in the main article

Reply: We agree on this and suggest to add text related to natural/synthetic $c$-$C_4F_8O$ in the main text of the manuscript, see replies to one of the above general comments by the reviewer. The key sentence there is: "Given these limitations, we are not able to draw any conclusions on any potential naturally-occurring $c$-$C_4F_8O$."

Reviewer Comment: Technical corrections:
p 5 l7 use indices in $c$-$C_4F_8O$
Reply: This is now fixed an the chemical formula is written with indices.

p 5 l11-14 repeats l4-8
Reply: Thank you for spotting this error, we plan to remove the second mentioning

p 5 l16 and 18 suppress ))
Reply: We believe that both closing parentheses are necessary.

p 6 l12 from observations of
Reply: Fixed, we removed the erroneous comma.

New references:
Reply: We have now added the Kočišek et al. (2018) reference, we did not need to add the Buizert et al. (2012) reference.